# Clinical Metagenomics Is Increasingly Accurate and Affordable to Detect Enteric Bacterial Pathogens in Stool

**DOI:** 10.3390/microorganisms10020441

**Published:** 2022-02-15

**Authors:** Christy-Lynn Peterson, David Alexander, Julie Chih-Yu Chen, Heather Adam, Matthew Walker, Jennifer Ali, Jessica Forbes, Eduardo Taboada, Dillon O. R. Barker, Morag Graham, Natalie Knox, Aleisha R. Reimer

**Affiliations:** 1National Microbiology Laboratory, Public Health Agency of Canada, Winnipeg, MB R3E 3R2, Canada; christy-lynn.peterson@phac-aspc.gc.ca (C.-L.P.); chih-yu.chen@phac-aspc.gc.ca (J.C.-Y.C.); matthew.walker@phac-aspc.gc.ca (M.W.); jennifer.ali@phac-aspc.gc.ca (J.A.); eduardo.taboada@phac-aspc.gc.ca (E.T.); dillon.barker@phac-aspc.gc.ca (D.O.R.B.); morag.graham@phac-aspc.gc.ca (M.G.); natalie.knox@phac-aspc.gc.ca (N.K.); 2Cadham Provincial Laboratory, Winnipeg, MB R3E 3J7, Canada; david.alexander@gov.mb.ca; 3Department of Medical Microbiology and Infectious Diseases, University of Manitoba, Winnipeg, MB R3E 0J9, Canada; hadam@sharedhealthmb.ca; 4Department of Biochemistry and Medical Genetics, University of Manitoba, Winnipeg, MB R3E 0J9, Canada; 5Shared Health, Winnipeg, MB R3A 1R9, Canada; 6Roy Romanow Provincial Laboratory, Regina, SK S4S 0A4, Canada; jessica.forbes@saskhealthauthority.ca; 7Department of Pathology and Laboratory Medicine, University of Saskatchewan, Saskatoon, SK S7N 0W8, Canada

**Keywords:** clinical metagenomics, pathogen detection, acute gastroenteritis, shotgun metagenomics, enteric

## Abstract

Stool culture is the gold standard method to diagnose enteric bacterial infections; however, many clinical laboratories are transitioning to syndromic multiplex PCR panels. PCR is rapid, accurate, and affordable, yet does not yield subtyping information critical for foodborne disease surveillance. A metagenomics-based stool testing approach could simultaneously provide diagnostic and public health information. Here, we evaluated shotgun metagenomics to assess the detection of common enteric bacterial pathogens in stool. We sequenced 304 stool specimens from 285 patients alongside routine diagnostic testing for *Salmonella* spp., *Campylobacter* spp., *Shigella* spp., and shiga-toxin producing *Escherichia coli*. Five analytical approaches were assessed for pathogen detection: microbiome profiling, Kraken2, MetaPhlAn, SRST2, and KAT-SECT. Among analysis tools and databases compared, KAT-SECT analysis provided the best sensitivity and specificity for all pathogens tested compared to culture (91.2% and 96.2%, respectively). Where metagenomics detected a pathogen in culture-negative specimens, standard PCR was positive 85% of the time. The cost of metagenomics is approaching the current combined cost of PCR, reflex culture, and whole genome sequencing for pathogen detection and subtyping. As cost, speed, and analytics for single-approach metagenomics improve, it may be more routinely applied in clinical and public health laboratories.

## 1. Introduction

What would be an ideal, modern approach for gastroenteritis diagnosis? For acute care clinical laboratories, the ideal approach must be rapid, cost-effective, and accurately detect infections (and co-infections) due to common, rare, and emerging enteric pathogens. Reference laboratories have additional requirements. Testing must provide actionable antimicrobial susceptibility information, as well as subtyping data to support identification, surveillance, and investigation of outbreaks caused by contaminated food or water.

Traditional, culture-based methods are cost-effective, but testing algorithms can be complicated, and turnaround times are typically more than 24 h. Although once considered the gold standard, it is now recognized that culture is less sensitive than molecular detection methods and can leave up to 80% of cases unresolved [1]. However, when successfully cultured, an isolate allows definitive pathogen identification and supports both susceptibility testing and outbreak investigation activities.

Culture-independent diagnostic tests, such as polymerase chain reaction (PCR)-based syndromic panels, have become increasingly popular. These assays are sensitive and turnaround times are measured in hours, not days. Per test costs vary, but many panels are compatible with automated testing platforms, which simplify workflow and reduce labor costs. However, commercial syndromic panels are optimized for detecting the most common enteric pathogens. Organisms not targeted by a given panel cannot be detected. Moreover, syndromic panels do not support susceptibility testing or subtyping activities for surveillance and thus, cannot fully replace culture and downstream workflows. As a result, recovery of an isolate should still be attempted for PCR-positive stools (termed reflex culture) to support food safety investigations and other public health surveillance activities [2,3].

A metagenomics-based approach can provide sensitive and specific pathogen detection as well as the necessary strain typing information required for critical public health and food safety activities [4]. To date, clinical diagnostic laboratories have not embraced metagenomics for routine testing. The expense, turn-around time, and analytical expertise required have been cited as barriers to routine use [5,6]. However, ongoing technological advancements in nucleic acid sequencing [7,8] and bioinformatics [9] are making metagenomics faster, less expensive, and more accessible. Proof-of-principle studies have highlighted the value of metagenomics for the clinical laboratory [7,8], including potential for pathogen detection in human stool specimens [10,11,12,13]. Stool metagenomes have been shown to support culture-independent strain typing [14,15,16] and single nucleotide polymorphism (SNP)-level differentiation of concurrent, epidemiologically unrelated, outbreaks [17]. Clusters of gastroenteritis caused by new or emerging viral pathogens have also been detected and investigated using metagenomics [18,19,20,21]. 

As a step towards implementation, we set out to compare the performance of culture, a PCR-based syndromic panel, and shotgun metagenomics to detect foodborne infections in stool. We also evaluated how different bioinformatics tools influence metagenome analyses. Through this work, we demonstrate that metagenomics can detect common enteric bacterial pathogens in stool with high sensitivity and specificity at a moderate cost. 

## 2. Materials and Methods

In this study, shotgun metagenomic sequencing was performed on a total of 304 human stool specimens from 285 patients with lab-confirmed gastroenteritis presenting to health care facilities in Manitoba, Canada. Testing methods are summarized in Figure 1. Specimens were collected by either diagnostic microbiology laboratories at Shared Health (SH; *n* = 130) or Cadham Provincial Laboratory (CPL; *n* = 174). 

Specimens collected by SH were processed for culture to detect *Salmonella* spp., *Campylobacter* spp., *Shigella* spp., or shiga-toxin producing *Escherichia coli* (STEC) O157. The growth of these organisms was supported using a combination of selenite broth (BD Difco™, BD Biosciences, Mississauga, ON, Canada), Cefixime Tellurite Sorbitol MacConkey agar (BD Difco™), MacConkey agar (Oxoid, Nepean, ON, Canada), Xylose Lysine Deoxycholate agar (BD Difco™), and *Campylobacter* selective agar (Oxoid, Nepean, ON, Canada). Organisms were identified by matrix-assisted laser desorption/ionization time of flight (MALDI-ToF; MALDI Biotyper^®^, Bruker Daltonics, Milton, ON, Canada) and VITEK^®^ 2 (bioMérieux, Saint-Laurent, QC, Canada). Of the 130 specimens collected by SH and included in this study, 86 were randomly selected for metagenomic sequencing from specimens submitted for culture over a one year period (June 2015–March 2016). Four of the randomly selected 86 were culture-positive. Of the remaining 44 specimens collected by SH, and included in this study, 41 were culture-positive for *Salmonella* spp., *Campylobacter* spp., *Shigella* spp., or STEC and three were culture-negative. 

Specimens collected by CPL were tested by both BD MAX Enteric Bacterial Panel (EBP; BD Biosciences, Mississauga, ON, Canada)—a multiplex PCR-based syndromic assay for detection of *Salmonella* spp., *Campylobacter* spp., *Shigella* spp., and STEC [22], and culture using a combination of selective and differential media, including selenite broth, Xylose Lysine Deoxycholate agar, Sorbitol-MacConkey agar, and Skirrow’s medium. The identity of cultured organisms was confirmed by MALDI-ToF (VITEK^®^ MS, bioMérieux, Saint-Laurent, QC, Canada) and VITEK^®^ 2 (bioMérieux, Saint-Laurent, QC, Canada). The specimens collected by CPL in 2017 were part of an unpublished validation study of more than 4000 specimens (both culture and EBP performed), while the specimens collected by CPL in 2019 were tested using EBP followed by culture on EBP-positive specimens (reflex culture). Among the 174 CPL specimens included in this study, 31 samples were taken from 12 patients an average of 1.4 days apart. Of the 174 CPL specimens, 96 were both EBP- and culture-positive, 8 were both EBP- and culture-negative, 66 were EBP- and culture-discordant, and 4 were EBP-indeterminate. After conventional testing, stool specimens were frozen at −80 °C and shipped on dry ice to the National Microbiology Laboratory where they were kept frozen until they could be processed for metagenomics sequencing.

For samples for which EBP was not performed (samples collected by SH), EBP was indeterminate, or EBP and culture results were discordant, additional testing with a second multiplex syndromic PCR panel, the BioFire FilmArray Gastrointestinal Panel (BF; bioMérieux Canada Inc., Saint-Laurent, QC, Canada), was performed at the Public Health Agency of Canada’s National Microbiology Laboratory (NML) (Winnipeg, MB, Canada). The laboratory performing the metagenome analyses (NML) was blinded to all diagnostic and culture results until preliminary metagenome analysis of the samples was complete.

A power analysis was performed to determine the required sample sizes to achieve the sensitivity and specificity to detect the pathogen of interest as previously described by Negida et al. [23]. Targeting a power value greater than 80% and a *p*-value less than 0.05, the minimum total sample sizes required to determine sensitivity and specificity were 190 and 192, respectively.

Specimens were stored at room temperature or 4 °C at the clinical laboratory for up to 7 days until they were transported to the NML. DNA was extracted within 24 h of arriving at the NML or stored at −80 °C until DNA extraction could be performed. DNA was extracted from 200 μL or approximately 0.2 g of each stool specimen using the QIAamp Fast DNA Stool minikit or the Zymo Research Fecal DNA kit and performed according to manufacturer’s instructions. The DNA concentrations were measured with the Qubit^®^ ds DNA High Sensitivity and Qubit^®^ dsDNA Broad range assay kits (Invitrogen, Burlington, ON, Canada). 

Illumina Nextera XT paired-end shotgun sequencing libraries were prepared from approximately one ng of DNA and sequenced using the MiSeq v3 600 cycle (2 × 300 bp), MiSeq v3 500 cycle (2 × 250 bp), or Nextseq 500/550 mid output 300 cycle (2 × 150 bp) kit protocols (Illumina Inc., San Diego CA, United States). Library size was determined using the Agilent Tape Station and normalization was performed manually. Library size selection was not performed. In situations where an extracted sample failed quality control parameters for library generation, a second extraction was performed using the DNeasy PowerSoil/Fecal kit (Qiagen Inc. Canada, Toronto, ON, Canada) to remove inhibitors. Paired-end, indexed reads were generated on either the Illumina MiSeq or NextSeq platforms according to manufacturer’s instructions. Sequencing libraries were either run individually or pooled. Sample-specific details are tabled in Appendix A. Two stool samples, where the amount of human DNA was high upon initial sequencing, underwent saponin-based host depletion as described previously by Charalampous et al. [24] and were re-sequenced.

A reference set of publicly available stool metagenomes from healthy donors (*n* = 6) were accessed from the short read archive (SRA), accession numbers: P23C0-SAMEA3182066 (ID: 3391602), P25C0-SAMEA3182067 (ID: 3391603), P38C0-SAMEA3182068 (ID: 3391604), P6C0-SAMEA3182050 (ID: 3391586), P7C0-SAMEA3182051 (ID: 3391587), and P8C0-SAMEA3182052 (ID:3391588) [25]. Raw sequence reads were stored on an in-house instance of IRIDA (https://www.irida.ca/, accessed on 13 December 2021) for data management. Quality filtering of reads was performed using FastQC (v0.72) [26], with subsequent adaptor and read trimming via Trimmomatic (v0.36.5) [22] (SLIDINGWINDOW = 20) on the Galaxy platform (v20.01) [27]. Deconseq (v0.4.3) [28], using in-house databases, was then used to remove contaminating sequences (human and PhiX genomic sequence) while retaining microbial and viral genomic reads (RefSeq genome sequences). The human database was generated from the GRCh38.p11 human genome assembly. The phiX database was constructed using accession number NC001422 (12 October 2017). The microbial and viral databases comprised of RefSeq pulls from the bacterial and viral repositories on 16 October 2017. As deconseq was designed for single-end data, the forward and reverse reads were analyzed independently, with paired-end files being restored thereafter using the FASTQ interlacer (v1.2.0.1) and deinterlacer (v1.1.2) tools on Galaxy [26]. Nonpareil (v3.1.1.0) [29] was used to determine the fraction of genomes recovered in each final quality-filtered and processed dataset to quantify the diversity of the community sampled and a given sequencing effort.

To obtain a microbial profile of the specimens, taxonomic abundances were estimated using Kraken2 (v2.1.1) [30] profiling data and Bracken (v1.0) [31] with read length set to 150 for the diseased cohort and 100 for healthy based on read lengths of these datasets. The abundance data was then normalized using the CSS (cumulative sum scaling) approach with a normalization factor of 0.9925 [32]. PCoA (principal coordinates analysis) plots were generated using the Bray–Curtis dissimilarity metric on taxa abundance across samples. Samples in the plots were then visualized and labelled according to their corresponding groups of variables. A non-parametric multivariate statistical test, PERMANOVA (permutational multivariate analysis of variance), was then conducted to compare groups using the Bray–Curtis dissimilarity as the distance measure between samples using R (v4.1.1; Vienna, Austria) [33] and the ‘vegan’ R Package (v2.5-7) [34]. Groups for culture, patient, DNA extraction kit, and sequencing platform variables were tested independently. Post-hoc analysis was conducted using the ‘pairwiseAdonis’ R package (v0.4) [35].

Based on our previous study [36], a taxa filter was implemented to retain species with a normalized abundance value of at least 250 in at least 60 samples to avoid spurious results from low-abundance taxa, which resulted in 72 taxa being reported. A sample filter was used to retain only the first sample from individuals with multiple samples. Differential abundance analyses were then conducted post-taxa filtering. Pair-wise comparisons between all diseased samples versus healthy were conducted using the Wilcoxon rank sum test, and multi-group comparisons were conducted using the Kruskal–Wallis test. *p*-values were adjusted using the post-hoc Benjamin Hochberg (BH) correction to address multiple hypothesis testing.

Four bioinformatics tools were assessed for pathogen detection: Kraken2 (v2.1.1) [30], MetaPhlAn (v2.6.0.0) [37], SRST2 (v0.3.7) [38], and KAT-SECT (2.3.2) [39]. KAT-SECT was run using default parameters. SRST2 also used default parameters, with the exception of the “MAXimum number of MISMATCHes per read for gene” allele calling increased to 150 to ensure overhanging mapped reads were not filtered out. Both tools used the following databases consisting of curated gene sequences: *Shigella*, *Salmonella,* and *Campylobacter* virulence factor databases (VFDB) (http://www.mgc.ac.cn/VFs (accessed on 17 March 2015) (545, 2547, 683 alleles respectively); and two in-house databases: (1) a 732-locus *Campylobacter jejuni* core genome multilocus sequence typing (cgMLST) scheme (3387 alleles total), and (2) *stx* alleles adapted from *Escherichia* VFDB and Flemming-Scheutz (145 alleles total). An allele was identified as present in a specimen based on default settings in SRST2, and a 10% non-zero kmer coverage cut-off was chosen for KAT-SECT as a first pass conservative threshold. MetaPhlAn and Kraken2 were run using default parameters. The MetaPhlAn standard database was packaged with the tool. For Kraken2, a standard database was created from a RefSeq pull on 26 November 2020 (kmer-len = 35, minimizer-len = 31, minimizer-spaces = 6, load-factor = 0.7). The following outputs were compared between tools: the sum of the number of reference database genes present in the read set of a sample based on criteria listed above (SRST2 and KAT-SECT); and the fragment percentage and relative abundance of reads that could be taxonomically classified (Kraken2 and MetaPhlAn, respectively).

A visual inspection of the tool comparison boxplots indicated that KAT-SECT provided the best separation between culture-negative and culture-positive samples. Thus, we chose to further evaluate KAT-SECT for pathogen detection and assess its analytical accuracy. K-mer coverage outputs were sorted in descending order and visualized along the *x*-axis. To determine the positivity threshold and therefore refine our pathogen detection methodology, we randomly selected a third of our samples (*n* = 104) to use as a training set. Specifically, this training set was used to determine the percent k-mer coverage (as determined using KAT-SECT) by which a sample is deemed ‘positive by clinical metagenomics’. The remaining 206 metagenomes were used as a validation data set to determine the sensitivity and specificity of pathogen detection. 

A composite standard including both culture and PCR was used to indicate the ‘true’ result. The composite standard was deemed positive if the specimen was culture-positive. In cases where culture was negative or not performed, the composite standard was deemed positive if two different PCR assays (EBP and BF) were both positive. The composite standard was deemed negative if negative by two of three conventional assays used in this study (EBP, BF, and culture). Publicly available metagenomes from six healthy individuals were assumed negative by culture and PCR. Receiver operator characteristic (ROC) curves for the training set were generated relative to the composite standard using KAT-SECT. The generation of (ROC) curves and determination of the optimal Youden’s index were previously described [40].

To determine the limit of detection, quality-filtered and host-removed metagenomes from culture-positive stool specimens (*n* = 146) were subsampled using seqtk_sample [40] to the following number of paired reads 10,000, 50,000, 100,000, 200,000, 1,000,000, and 5,000,000 using unique random number generator (RNG) seeds. Specimens with lower read amounts upon initial sequencing were subsampled to the maximum level allowed from native read counts. Pathogen detection in subsampled read sets was conducted using KAT-SECT as described above. Relative abundances were calculated as the number of genes that had greater than 1% non-zero k-mer coverage (0.5% for Campylobacter) divided by the total number of genes in the queried database.

All outputs from KAT-SECT, SRST2, Metaphlan, Kraken2, and Nonpareil were analyzed using custom R (v4.1.1; Vienna, Austria) scripts [33], written in RStudio (v1.4.1717; Boston, MA, USA) to generate the Tables and Figures presented. Scripts used the following R packages: here (v1.0.1) [41] and readr (v2.0.1) for file pathway handling and reading of data files; tidyr (v1.1.3), dplyr (v1.0.7), stringr (v1.4.0), and purrr (v0.3.4) for data wrangling and tidying; and ggplot2 (v3.3.5) to create plots [42].

## 3. Results and Discussion

### 3.1. Samples and Conventional Results

#### 3.1.1. Sample Provenance

We sequenced the metagenomes of a total of 304 stool specimens from 285 patients presenting with gastroenteritis. Specimens were tested by culture and/or PCR for *Campylobacter* spp., *Salmonella* spp., *Shigella* spp., and STEC as part of routine clinical testing (Appendix A). Of the 86 specimens randomly collected over a one year period, four (5%) were culture-positive (three *Salmonella enterica*, one *Campylobacter jejuni*). 

#### 3.1.2. Clinical Accuracy of PCR Compared to Culture

The sensitivity and specificity of PCR compared to culture were 97% and 89%, respectively (Appendix A). Among 260 stool specimens tested by both culture and PCR, 73 (28%) were discordant for at least one of the four pathogens tested (Appendix A). Of these, 18 were excluded from further culture–PCR comparisons due to BF-EBP discordant results. Among the remaining 55 samples where PCR and culture were discordant, most (91%) were PCR-positive but culture-negative. A total of 21 out of 55 culture-PCR discordant samples (38%) were PCR-positive for STEC but culture-negative—in most of these cases (18 of 21 = 86%) metagenomics agreed with culture. Of the 55 culture-PCR discordant results, 17 were PCR-positive (31%) for *Campylobacter* spp. but culture-negative, whereas 10 of those 17 were both culture and metagenomics negative. A total of 10 out of 55 culture-PCR discordant samples were PCR-positive for *Shigella* spp. but culture-negative, and of those, metagenomics agreed with culture in five (50%) instances. Of the 55 culture-PCR discordant results, 10 were PCR-positive (18%) for *Shigella* spp. but culture-negative, whereas 5 of those 10 were both culture and metagenomics negative. Taken together, these results suggest that either the sensitivity of culture and metagenomics are limited or PCR suffers from target specificity issues (Appendix A). While culture is inherently limited to detect live organisms, culture practices may also suffer from sensitivity losses due to the fastidious nature of some organisms, particularly *Campylobacter* spp. and STEC, inhibitory growth of competing genera, and a viable but not culturable state [43]. Of relevance, a proportion of samples included in this study were specifically selected for metagenomics sequencing because they were culture- and EBP-discrepant among a larger group of samples tested during a 2017 validation study with routine microbiological testing (unpublished, personal communication). Thus, it is important to note that the discrepancy we observed in this study is not an accurate measurement of the true incidence of discrepancy between culture and PCR.

### 3.2. Microbial Community Profiling

To assess the abundance of potential pathogens in each specimen, we taxonomically classified reads from 304 metagenomes from 285 cases of gastroenteritis and six stool metagenomics from healthy donors [25] using Kraken2 [30]. The majority of reads acquired for each specimen was of bacterial origin (average 52.7%; interquartile range (IQR) 36.9–70.8%). In specimens where *Campylobacter* spp., *Salmonella* spp., and *Shigella* spp. were detected by culture, an average of 0.897% (IQR 0.040–0.43%), 5.60% (IQR 0.083–1.89%), and 4.43% (IQR 0.045–1.85%) of reads corresponded to that pathogen, respectively. For specimens that tested positive for the *stx* gene by culture and/or PCR, an average of 5.60% (IQR 0.088–2.57%) of reads corresponded to *E. coli*. Overall, an average of 3.44% of reads (IQR 0.048–1.45%) corresponded to the pathogen of interest. In previous studies, the relative abundances of bacterial pathogens in stool ranged from 0.02% to 45.4% [12,13]. This is in contrast to other human specimen types where, without a host depletion step, the pathogen genome can comprise as low as 0.0012% of reads [44,45]. 

To assess the microbial community composition, reads from the 304 diarrheal specimens and six stool metagenomes from healthy donors, Kraken2 results were fed into Bracken to estimate relative abundances, and normalized using the CSS method. The average relative abundance of the genera of interest in each group are depicted in Figure 2. Of note, *Shigella* spp. and *Campylobacter* spp. were detected in several stool specimens that were culture-negative for these pathogens. This specificity issue could be the result of non-specific sequences in the database queried or represent pathogens that were truly present but not culturable.

To explore the potential of community profiling for pathogen detection, we first assessed the similarity of metagenomes based on pathogen/disease state (culture result), patient, extraction kit, and sequencing platform variables. We performed PCoA using the Bray–Curtis dissimilarity distance measure among trimmed and normalized metagenomes, and conducted the PERMANOVA test on each variable for differences in centroid and dispersion among groups (Figure 3 and Appendix A). If metagenomes containing the same pathogen group together, and distinct, from metagenomes containing other pathogens, then community profile could be explored as a potential pathogen detection tool. While visual observation of PCoA plots shows clustering of healthy stool metagenomes, but not diseased, all variables were significant with adjusted *p*-values ≤ 0.003, and post-hoc analyses showed pair-wise comparisons between groups within each variable (Appendix A). While visually scattered on the PCoA plot, centroids measured for each pathogen differed significantly (*p* = 0.001), and significant differences were highlighted by comparisons of diseased to healthy samples. The centroids representing different sequencing platforms as well as DNA extraction protocols employed in this project were also significantly different (*p* = 0.002, 0.001), indicating that standardized collection, storage, extraction methodologies, and sequencing platform would need to be in place to optimize pathogen detection based on microbiome analysis. While the difference between metagenomes containing distinct pathogens was statistically significant, the distinction would not be sufficient to categorize metagenomes into pathogen types. As a result, our initial microbial profiling results do not support the use of community profiling for pathogen detection and thus we did not explore this further.

Previously, Chen and Tyler [36] found that due to the differences in the limit of detection and sensitivity between protocols, taxa with higher abundance and prevalence exhibited more consistent abundance levels between data from experimental protocols. Therefore, we proceeded to conduct the differential abundance test on 72 taxa with higher abundance using a stringent taxa filter.

To explore the possibility of leveraging microbiome data as a rapid screen for foodborne outbreak detection, we compared stool metagenomes taken from patients at different time points during their infection. If PCoA could not identify metagenomes from the same patient, then it may be excluded as a potential method to detect ‘matching’ microbiomes from patients who consumed the same contaminated food (Appendix A). Twelve patients had more than one stool sample sequenced in this study. Post-hoc analysis did not reveal significant differences between paired individuals, although the small sample size (two to three specimens per patient) limits the statistical power and any conclusions we may draw. Our results do not support using community profiling as a rapid screen for outbreak detection. However, in a previous study, microbial communities from two outbreaks were distinct from each other and distinct from healthy samples based on non-metric multidimensional scaling (NMDS) of normalized composition at the genus level [14].

To assess differential abundance between gastroenteritis and healthy samples, taxa associated with disease state and pathogen type were tested using the Wilcoxon rank sum test at the species level on the filtered dataset (Appendix A). In this analysis, specimens from patients presenting gastroenteritis, irrespective of etiology, demonstrated a significantly higher abundance of *E. coli* and *Klebsiella pneumoniae* compared to healthy controls (BH-adjusted *p* = 0.0358 and 0.0267, respectively). The abundance of *E. coli* present in STEC infections was not significantly different compared to gastroenteritis caused by other bacterial pathogens (Figure 2 and Appendix A). This agrees with the previously observed increase in the abundance of *E. coli* during infections with *Salmonella* spp., norovirus, and other enteric pathogens [17,46,47]. In the healthy stool specimens analyzed, species, such as *Hippea maritima*, *Acidithiobacillus caldus*, and *Phocaeicola vulgatus* showed higher abundance levels in the healthy group (adjusted *p* = 0.0198). Of the three, *Hippea* and *Acidithiobacillus* are environmental microbes and may represent contaminants. *Phacaeicola* is not well known but is part of the bacteroidetes phylum, common to the gut microbiome. We further conducted the Kruskal–Wallis test to identify genera with significant differential abundance among culture groups and healthy. In order of significance, genera, including *Klebsiella*, *Leadbetterella*, *Salmonella*, *Bacteroides*, *Cystobacter*, *Phocaeicola*, *Sulfurimonas*, and *Tannerella* were significantly differentially abundant among cohorts (adjusted-*p* < 0.05) (Appendix A). All but *Klebsiella* were of higher abundance among the healthy samples.

### 3.3. Pathogen Detection Using Metagenomics

#### 3.3.1. Tool and Method Comparison

In most studies that have applied a metagenomics approach to pathogen detection, both the sensitivity and specificity of this assay fared poorly against existing methods, such as culture and PCR [15]. In a 2015 comparison of different tools for taxonomic classification, a wide range of variability was found in both sensitivity and overall accuracy [48]. This study tested four analysis tools (Kraken2, KAT-SECT, MetaPhlAn, and SRST2) to determine the best candidate to implement into our protocol for routine sequencing of stool specimens at our reference laboratory (Figure 4). Informatics tools designed for marker detection (KAT-SECT, SRST2) and databases containing only pathogen-specific targets provided better sensitivity and specificity than tools and databases designed for taxonomic classification (Kraken2 and MetaPhlAn). For example, only KAT-SECT (Figure 4A) and SRST2 (Figure 4B) could uniquely distinguish STEC, whereas Kraken2 (Figure 4C) and MetaPhlAn (Figure 4D) detected generic *E. coli*. As introduced in Section 3.2 (Figure 2), Kraken2 detected *Shigella* spp. and *Campylobacter* spp. in several specimens that were culture-negative for these organisms. In some cases, these organisms were detected by all four tools suggesting that these organisms were truly present but were not culturable as is common for molecular-based tests. Overall, KAT-SECT combined with pathogen-specific marker databases was markedly more accurate than SRST2 and classifier databases. Using KAT-SECT, metagenomes from culture-positive and culture-negative stool samples are distinguishable based on the percentage of target alleles covered by k-mers (Figure 5). As such, KAT-SECT with pathogen-specific marker databases was used for downstream validation of this approach in our reference laboratory.

#### 3.3.2. Training Set

To refine our pathogen detection methodology, we randomly selected a third of our samples (*n* = 104) as a training set. Specifically, this training set was used to determine the percent k-mer coverage (as determined using KAT-SECT) by which a sample is deemed positive using clinical metagenomics. The remaining 206 metagenome sequences were used as a validation set to determine the clinical accuracy of pathogen detection. 

Receiver operator characteristic curves for the training set were generated relative to the composite standard using KAT-SECT (Appendix A). The optimal Youden’s indices were 1.0% (80th allele), 1.0% (50th allele), 1.0% (180th allele), and 0.5% (top allele) for *Salmonella*, *Campylobacter*, *Shigella,* and *stx*-gene carrying organisms, respectively.

#### 3.3.3. Sensitivity and Specificity of Metagenomics

The overall accuracy of clinical metagenomics, measured in the validation set, was comparable to culture with a sensitivity and specificity of 91.2% and 96.2%, respectively (Appendix A). Among all samples that were tested by both culture and metagenomics, we detected a pathogen in the metagenomes of 20 culture-negative samples (2 *Salmonella* spp., 3 *stx*-gene carrying organisms, 5 *Shigella* spp., and 10 *Campylobacter* spp.); however, all but three of those were positive by PCR indicating that metagenomics detected the pathogen with greater sensitivity than culture and that the true specificity of our metagenomics approach is closer to 100%. Exactly half (10 of 20) of the culture-negative, PCR-positive specimens indicated the presence of *Campylobacter* spp. using both PCR and metagenomics. This is consistent with the fastidious nature and difficulty in recovering *Campylobacter* spp. from stool and the resulting poor sensitivity of culture. A *Campylobacter* spp. isolate was recovered in a single stool sample, but both PCR and metagenomics were negative for *Campylobacter*, instead finding *Salmonella enterica*. We expected the integrity of *Campylobacter* spp. DNA to limit the sensitivity of PCR and metagenomics; however, in this study, we observed higher sensitivity for the detection of *Campylobacter* spp. compared to other pathogens.

The sensitivity and specificity of metagenomics against PCR were 76.5% and 99.8%, respectively (Appendix A). The overall accuracy of metagenomics when measured against PCR was most affected by poor sensitivity to detect *stx* compared to PCR. In 23 of 248 specimens (9.3%) that received PCR (excluding samples where EBP and BF conflicted), PCR detected *stx* when metagenomics did not. Most (18 of 23) of these cases were culture-negative. A lower sensitivity is not unexpected as the STX database consists of alleles for one marker gene versus a database containing many marker genes for the other pathogens and therefore suffers in cases of lower sequencing depth.

Specimens where culture and EBP results disagreed (or where EBP results were indeterminate) were re-tested using a second commercially available multiplex syndromic PCR panel (BF). Of 78 samples that received both EBP and BF, 22 (28%) were discordant for the detection of one or more pathogens. Among 22 discordant samples, the discordance was observed in the detection for *Campylobacter* spp., *Salmonella* spp., *Shigella* spp., and STEC, in 16 (73%), 2 (9%), 3 (14%), and 3 (14%) cases, respectively. Of note, two samples were discordant for both the detection of *Shigella* spp. and STEC. Most of the discrepancy between the two multiplex PCR panels was for *Campylobacter* spp., which may be due to the number of *Campylobacter* spp. targeted by each assay; both the EBP and BF panels detect *C. jejuni* and *C. coli*, while the BF panel also detects *C. upsaliensis*. However, in the 16 samples discordant for *Campylobacter* spp., BF was negative 75% (12/16) of the time. Congruently, in three of the four cases where *Campylobacter* spp. was detected with BF but not EBP, metagenomic sequencing identified *C. upsaliensis* (Figure 5). For *Salmonella* spp., the two conflicting PCR results were both positive using the EBP yet negative using BF. For *Shigella*, all (3 of 3) conflicting PCR results were EBP-positive yet BF-negative. These results indicate either higher sensitivity or lower specificity of EBP compared to BF. However, all (3 of 3) conflicting *stx* PCR results were BF-positive and EBP-negative, suggesting higher sensitivity or lower specificity of BF compared to EBP for the detection of STEC. EBP was performed on fresh stool specimens, whereas BF was performed on frozen and thawed stools, which may explain some of the discordance. However, a previous study found that 66.6% of stool samples with discordant culture and BF results were confirmed with EBP, which is consistent with our observations [49].

The sensitivity and specificity of metagenomics against the composite standard were 79.7% and 99.8% overall, respectively. The sensitivities measured in our study are likely a conservative estimate of the true sensitivity of metagenomics as 55 of the specimens sequenced were selected based on discrepant culture and PCR results. However, it is important to include these specimens as widely used syndromic multiplex PCR panels are not always concordant [49]. 

#### 3.3.4. Factors Affecting Sensitivity of Metagenomics

The primary influencer of metagenomics sensitivity was depth of sequencing read coverage. In order to measure the extent of microbial diversity sampled, we assessed the redundancy of taxa present for specimens with varying depths of coverage (Appendix A). The coverage measurement, indicating the proportion of diversity sampled, also gives us an idea of the likelihood of recovering the pathogen at a given sequencing depth. Nonpareil was used to assess community complexity in 304 stool specimens sequenced. After host sequence removal using deconseq, 4 of 304 specimens had insufficient dehosted reads to run nonpareil. For the remaining samples, nonpareil predicted that an IQR of 0.36 to 2.11 GB of microbial sequencing data is required to recover 95% of organisms present in each sample. We sequenced an average of 0.779 GB of the non-human fraction of each specimen. Rodgriguez and Konstantinididis [50] found that 5.6 GB of sequencing effort is required to cover 97% of organisms present in a human single stool sample, although they did not remove host reads before their measurement. 

The presence of human DNA in some samples limits the sensitivity of metagenomics. An average of 13.8% (IQR 0.130–5.94%) of the metagenomes in this study were made up of human DNA reads, based on taxonomic classification using Kraken2. In our study, the visible presence of blood was a strong indicator of high human DNA content. A study by Pena-Gonzalez [47] found that mucous was the more significant predictor of human DNA level in stool samples; however, we found the presence of mucous was difficult to assess as specimens had been frozen prior to noting these observations. Samples that were composite-positive but metagenomics-negative (i.e., false negatives) had on average 40% more human DNA compared to all other samples tested.

DNA sequences mapping to the human genome were removed from the datasets prior to nonpareil and limit of detection analyses (Figure 1). An average of 6.7% of reads mapping to the human genome also mapped to Kraken’s microbial database. Cross-mapped reads (reads mapping to both the human genome and Kraken2′s microbial database) were retained. 

The conditions under which samples were stored prior to sequencing could have impacted results. Samples were frozen at −80 °C once received in the research laboratory but the temperature and duration of storage prior to arriving was inconsistent for the samples used in this study. In the days or weeks between sample collection and sequencing, it is possible that some organisms were lost or overgrown with other organisms. It is also possible that the high abundance of some pathogens, particularly *Salmonella* spp., in some samples was the result of growth during storage. For example, storage of stool samples at room temperature may increase the abundance of *Enterobacteriaceae* (now *Enterobacterales*) [51]; however, it is unknown if this applies to other enteric pathogens, such as *Campylobacter* spp. Storage conditions that enrich enteric pathogens in stool should be investigated further as a method to increase the sensitivity of pathogen detection.

#### 3.3.5. Factors Affecting Specificity of Metagenomics

We did not observe specificity issues in the detection of *Salmonella* spp. but Doster et al. [52] found that false positives were created when plasmid sequences were included with the bacterial chromosome in genome data when trying to identify *Salmonella* spp. in cattle feces. Our study indicates that this is alleviated by using targeted and curated databases, such as those containing only genus-specific core genes or virulence factors.

Clinical metagenomics using the KAT-SECT approach against a *Campylobacter* cgMLST database detected *C. upsaliensis* in specimens that were missed by PCR and/or culture (Figure 4). The detection of *C. upsaliensis* was only observed when using a database of core genes but not when using a *Campylobacter*-specific virulence factor database [53]. 

#### 3.3.6. Limit of Detection

A major limitation of metagenomics as a diagnostic tool is the often significant sequencing read coverage (and thereby cost) required to detect and confidently characterize the pathogen genome of interest. The proportion of pathogen present in a clinical metagenome is often very low. In the stool metagenomes studied here, an average of 3.44% of the sequence acquired was derived from the pathogen of interest (as detected by Kraken2). To better understand the lowest number of dehosted reads required from human stool, we subsampled the non-human fraction of each of our culture-positive stool metagenomes to contain 10,000, 50,000, 100,000, 200,000, 500,000, 1 million, and 5 million read pairs (Figure 6). The limit of detection (LOD) for enteric bacterial pathogens was low (<100,000 read pairs) in the majority of cases. To maximize sensitivity and specificity, an average of 125,000 read pairs were required to detect and identify the enteric bacterial pathogen isolated.

In this study, we observed 10 culture-positive samples where increasing the sequencing depth to an entire MiSeq run (13.2–15 Gb) did not yield evidence of the pathogen. Thus, we asked if there are limits to increasing the sequencing depth with a corresponding increase in sensitivity as compared against culture. In other words, is there a point at which more sequencing is not going to increase sensitivity? We found that the sensitivity of metagenomics increases steadily until 1–5 million read pairs, after which increasing the sequencing effort has a smaller impact on improved sensitivity (Figure 7). Other studies have shown that increasing coverage beyond a threshold will decrease specificity, although we did not observe that effect [30]. 

### 3.4. Practical Considerations

A major obstacle limiting the clinical application of shotgun metagenomics has been the sequencing read depth required to detect microbial pathogens amongst the often high background sequences from the host and/or other microbial flora. To achieve 90% sensitivity against culture (i.e., the amount of sequencing read data to yield 90% sensitivity against culture), the in-house cost of sequencing consumables ranged from $124–$519 (CAD)/stool specimen depending on the sequencing kit and platform used. This wide range in cost per sample is partly reflective of sequencing technology output increases and resulting cost decreases observed over the course of the project (2014–2019). Including DNA extraction and host depletion, our cost to extract and sequence 5 million 150 bp microbial read pairs (1.5 GB) was as low as $145 CAD/stool specimen. Currently, the cost of PCR (EBP) to detect four common bacterial enteric pathogens is $25 (CAD)/test. Expanding detection to a total of eight bacterial enteric pathogens using the extended EBP costs an additional $20 CAD/test. In routine practice, PCR-positive specimens are reflex cultured and recovered isolates undergo serotyping and/or whole genome sequencing (WGS) for a total cost of approximately $166 CAD per isolate [54]. At this cost, only four enteric bacterial pathogens are covered. Therefore, with modest improvements, the combined utility of metagenomics as both a diagnostic and public health tool could represent a modest cost savings overall and even a streamlined workflow. That said, the metagenomic sequencing depth required to perform high-resolution strain typing of pathogens in stool is not well established and will be greater than that required for pathogen detection. Encouragingly, a study by Huang et al. [17] reported as low as 0.5 GB of sequencing coverage to successfully distinguish two foodborne outbreaks caused by genetically similar *Salmonella enterica* subspp. *enterica* Serovar Heidelberg strains. While in this study we focused on detecting bacterial enteric pathogens, there is potential to further leverage metagenome sequence results for detection of other DNA-based organisms and in silico antimicrobial resistance (AMR) prediction, which would result in further cost efficiencies. 

Viewing the clinical and public health laboratory and epidemiology picture holistically, metagenomics could be more cost-effective once standardized and implemented than existing systems. Indeed, the implementation of newer technologies has broader cost impacts than the cost per test alone. For example, WGS has reduced the number of futile epidemiological investigations by focusing human resources on true outbreaks [55]. Secondly, the implementation of syndromic PCR panels has eliminated selective algorithms and complicated lab workflows, thereby reducing staffing requirements in both hours and expertise required. 

In this study, we employed a saponin-based host depletion step on select samples based on the presence of blood or initial sequencing results [24]. To further reduce the overall cost and speed of metagenomics, a host depletion step could be applied to all samples either at the time of DNA extraction prior to or during sequencing. A “read until” or adaptive sequencing approach is a promising new technique undergoing validation studies, that if applied to the detection of pathogens in stool would avoid sequencing human, plant, and animal foodstuffs, and normal microbiota [8].

Practical barriers of shotgun metagenomics for routine testing remain. Drawbacks to clinical metagenomics have been the high cost, slow turnaround-time (TAT), limited informatics tools, complexities of interpretation, and difficulties in diagnostic assay validation [21]. The future of metagenomics in the clinical laboratory will employ solutions, such as batched sequencing runs, including other pathogens and specimen types, employing long read sequencing with real-time base-calling and analytics [24,56], integrating host depletion steps for candidate specimens [24], and the adoption of a user-friendly analytical framework with simplified interpretation and reporting. Encouragingly, Charalampous et al. [24] reduced the TAT of metagenomics to less than six hours from sample receipt to reporting of pathogen identification and antimicrobial susceptibility results. However, the referenced study used a different sequencing platform than the one employed in this study. While the analytical tools and methods to detect pathogens in clinical specimens are well developed as shown in this study, future efforts must be dedicated to bridging sequencing data and results in a clear and interpretable fashion for uptake in point-of-care diagnostics.

## 4. Conclusions

Considering that the vast proportion of sequence acquired from a clinical specimen is not derived from the pathogen of interest, there is still a considerable cost to metagenomics sequencing. However, the frequently low limit of detection measured in this study is promising in the pursuit to reduce the cost of metagenomics, at least for stool specimens, where the pathogen is typically in relatively high abundance. Whereas pathogen-specific marker-based approaches outperform taxonomic classifier methods to identify common pathogens, classifier methods may still have a role to play in cases with rare or emerging enteric pathogens where the taxonomy is not well represented in the databases queried in a marker-based approach. Thus, metagenomics will still play an important role in reference laboratories, for the diagnosis and treatment of unresolving or complex cases, until technological advances improve the timeliness and cost of metagenomics for routine application. A future where both clinical and public health information are extractable from metagenomes will further increase the value and relevance of metagenomics approaches.

## Figures and Tables

**Figure 1 microorganisms-10-00441-f001:**
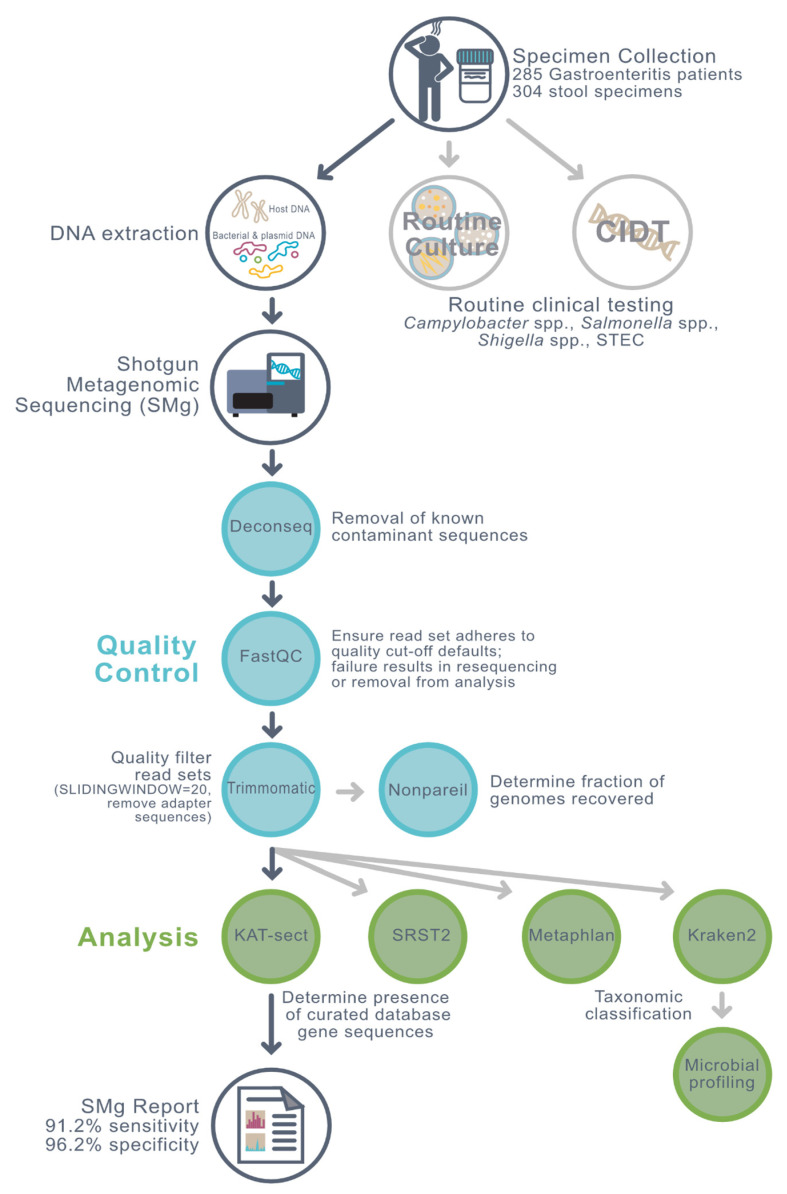
Stool specimens from patients presenting with gastroenteritis underwent culture, PCR, and shotgun metagenomics sequencing on an Illumina platform. Host sequences were removed using deconseq prior to organism coverage estimation using nonpareil and limit of detection measurement on trimmed reads having passed read quality control. Pathogens were detected in trimmed reads using four analytic tools followed by microbial profiling using normalized reads.

**Figure 2 microorganisms-10-00441-f002:**
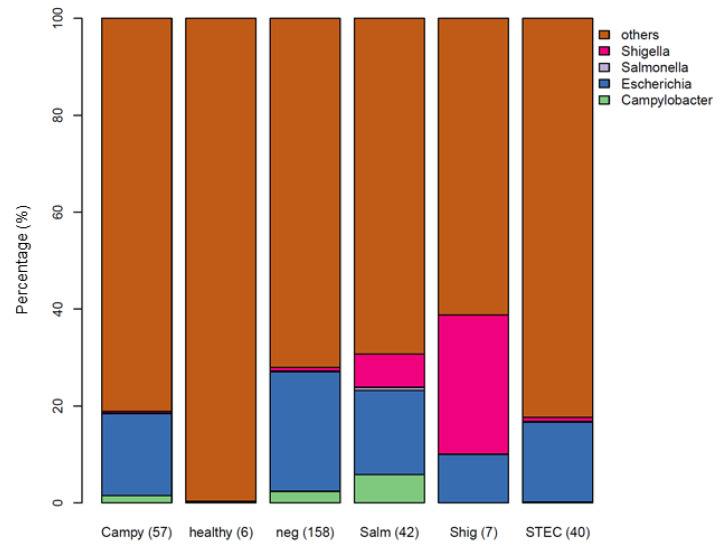
Relative abundance of bacterial genera detected in stool specimens, categorized by culture result. Reads were classified using Kraken2 and the abundance was estimated using Bracken. The number of samples in each category are shown in parentheses.

**Figure 3 microorganisms-10-00441-f003:**
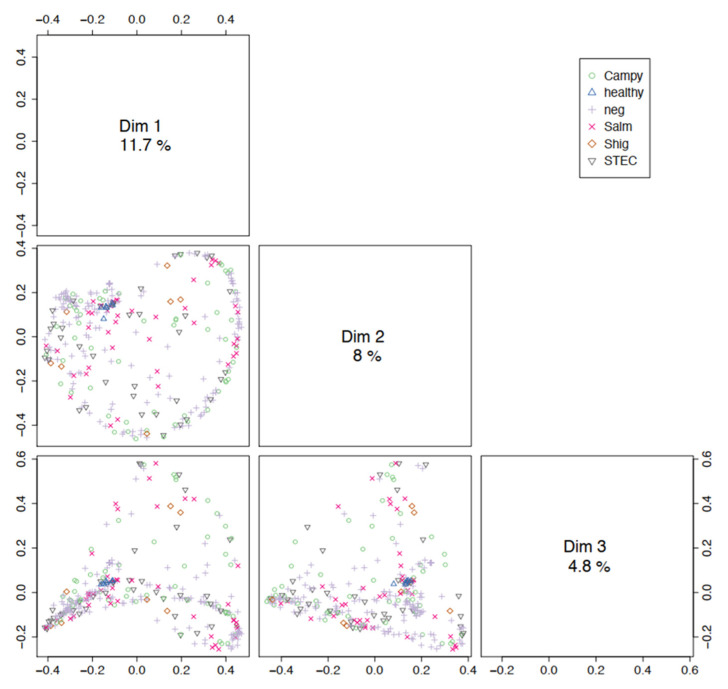
Principal coordinate analysis (PCoA) using the Bray–Curtis dissimilarity distance measure among trimmed and normalized metagenomes on groups within each variable, colored by culture result (with stringent filtering). Axes for each dimension comparison are labeled Dim 1, Dim 2, and Dim 3. Values plotted are top eigenvectors for each dimension. Percentages correspond to the total variance explained for that dimension.

**Figure 4 microorganisms-10-00441-f004:**
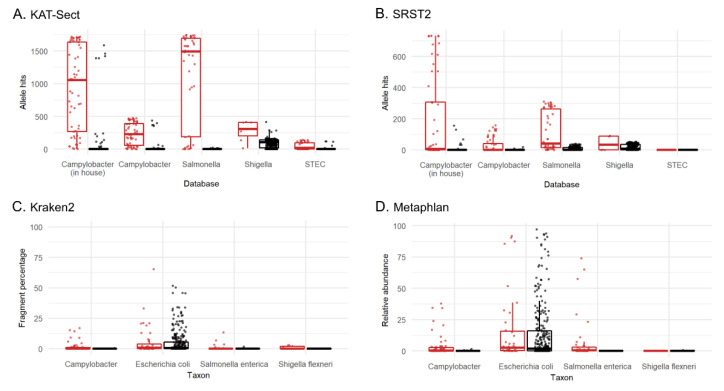
Pathogen detection tools compared: (**A**) KAT-SECT; (**B**) SRST2; (**C**) Kraken2; (**D**) Metaphlan. Culture-positive (red), culture-negative (black). *x*-axis displays the curated database chosen for each pathogen for KAT-SECT and SRST2 and the taxon for Metaphlan and Kraken2. Y-values plotted for KAT-SECT and SRST2 are the sum of the identified alleles from reference databases. An “allele hit” was defined by default settings in SRST2 and when k-mer coverage was greater than 10% in KAT-SECT. Values plotted for Kraken2 and MetaPhlAn are the percentage of fragments covered by the clade rooted at this taxon and the relative abundances of reads classified to species, respectively.

**Figure 5 microorganisms-10-00441-f005:**
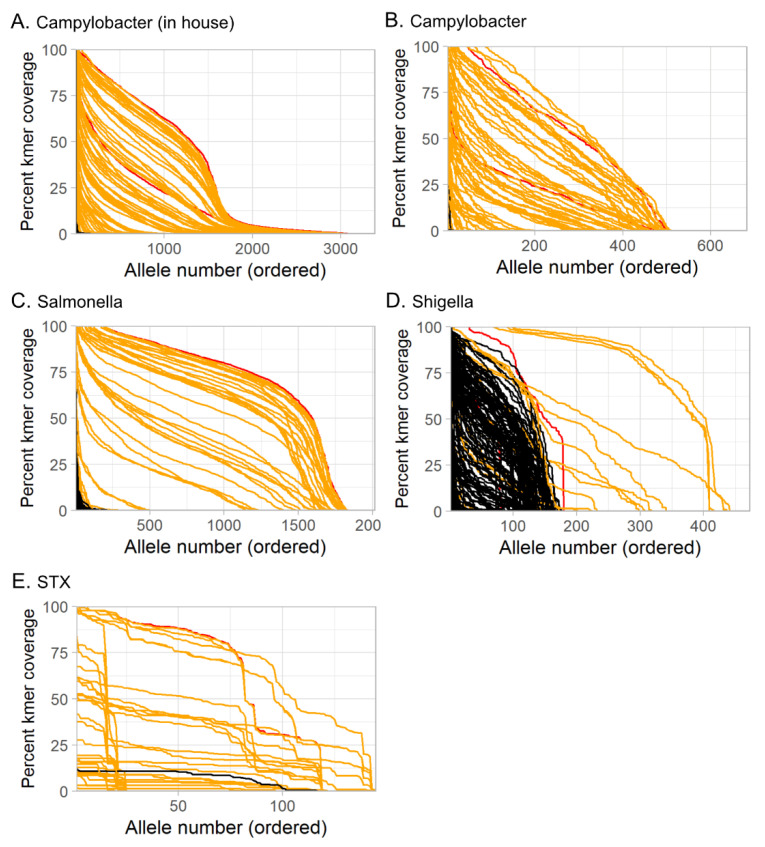
KAT-SECT identified alleles from curated databases for each pathogen of interest: (**A**) in-house campylobacter database; (**B**) Campylobacter VFDB database; (**C**) Salmonella VFDB database; (**D**) Shigella VFDB database; (**E**) STX in-house database. Whole genome sequences from pure culture isolates (red), and metagenomes from bacterial culture-positive (orange) and culture-negative (black) stool specimens. *x*-axis, the k-mer coverage against each allele in the database, sorted in descending order (allele rank number on *x*-axis). *y*-axis, percent of allele bases with at least 1-times kmer coverage.

**Figure 6 microorganisms-10-00441-f006:**
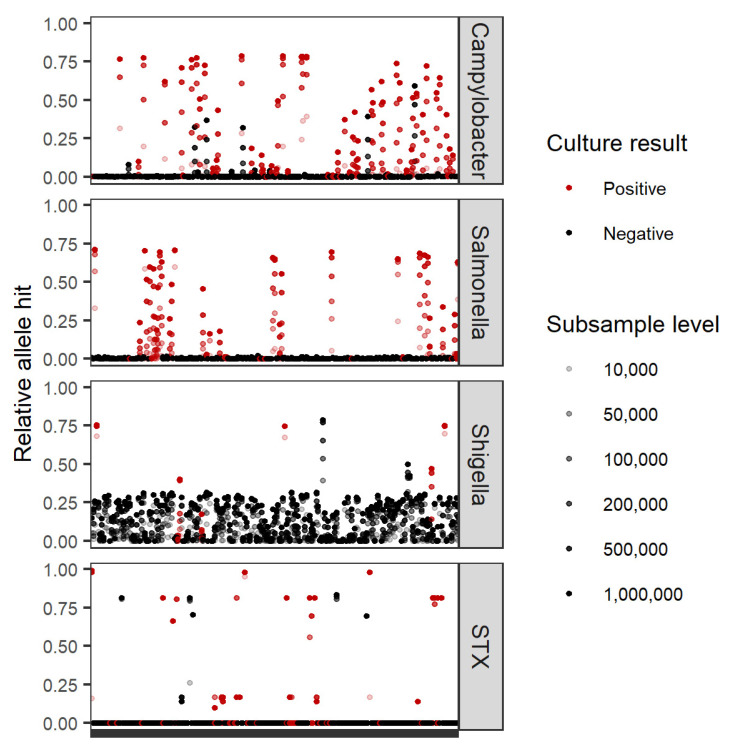
Relative allele hits of *Campylobacter* in house, STX, *Salmonella*., and *Shigella* databases in subsampled read sets to assess the limit of pathogen detection. Specimen read sets are plotted in alphabetical order on the *x*-axis. Relative allele hits are based on KAT-SECT analysis and calculated as total number of alleles that had greater than 1% (0.5% for *Campylobacter*) k-mer coverage divided by the total number of allele sequences in the database queried (*y*-axis).

**Figure 7 microorganisms-10-00441-f007:**
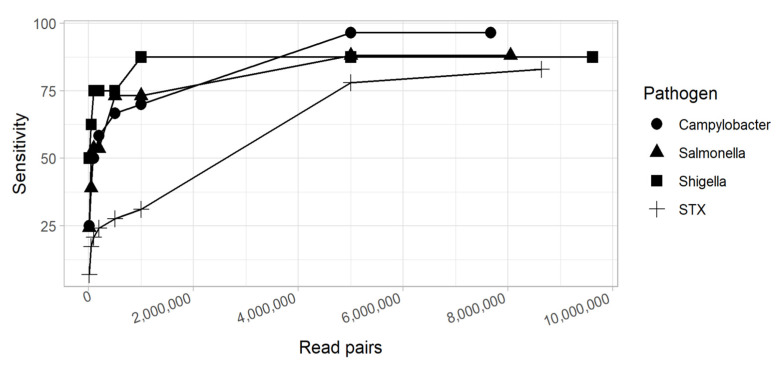
The microbial fractions from culture-positive datasets were subsampled and the percent sensitivity (*y*-axis) was measured at each subsample level (*x*-axis).

## Data Availability

The non-human fractions of the raw datasets presented in this study were deposited in the U.S. National Center for Biotechnology Information’s Sequence Read Archive in bioproject number PRJNA786578.

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
