# Peer review of "Clinical Metagenomics Is Increasingly Accurate and Affordable to Detect Enteric Bacterial Pathogens in Stool"

_microorganisms, 2022, doi:10.3390/microorganisms10020441_

Round 1

Reviewer 1 Report

The authors present a work about the evaluations of shotgun metagnomics for pathogen detection in stool. The experimental datasets are quite comprehensive, including culture, PCR, sequencing, MALDI-ToF. However, the presentation of this work was not well organized. The authors would like to evaluate five analytical approaches for pathogen detection: microbiome profiling by Kraken2, Kraken2, MetaPhlAn, SRST2, and KAT-SECT. However, the authors did not describe how to determine the presence of pathogen by using microbial composition. As shown in Figure 2, some Shigella and Campylobacter were detected in neg samples. Although healthy stool metagnomes are clustered in Figure 3, other pathogen samples seem scatter everywhere. I don't think it is a good way to present PCoA for the purpose of pathogen detection. The authors further used twelve patients had more than one stool sample sequenced in this study. However, their result did not support using community profiling as a rapid screen for outbreak detection. I would suggest the authors describe how they use microbial composition to detect pathogen and clearly show the result of "Mg and composite" (as shown in Dataset2). As for the comparison of the other four tools, the main result is shown in Figure 4 (with invisible x- and y-axis). However, the authors did not describe how to determine the cutoff values (of gene presence / of relative abundance) for pathogen detection. To my knowledge, Kraken2 and MetaPhlAn are two microbial profiling tools, while KAT is a k-mer analysis toolkit, KAT-SECT enables coverage estimation across sequences. How these tools can be used for pathogen detection? The authors should provide more descriptions. If I understand correctly, the authors used databases described in line 197-200 as the subject sequences of KAT-SECT. What are they? genome or gene sequences? how many sequences in each database? a different database results in different cutoff (Figure 4)? It is unclear how to produce Figure 5 and what can we get from this figure without visible values in x- and y-axis.

In the line 379-380, the training set was used to determine the percent k-mer coverage, however, I can not find any result that can be used to determine this value of 10%. 

As described in line 355-357, the authors would like to determine the best candidate protocol for reference laboratory, I would suggest the authors focus on how to do and present the results clearly.

Figure 6, Relative abundance of genes? or relative gene abundance ? this should be used in figure legend. What's the cutoff value for relative gene abundance? greater than 0 for Camppylobacter, Ecoli, Salmonella? or something else? 

Missing figure legend in Figure 7. What are the four symbols?

Line 547-598 to me, feel like the content of a review article. Another sequencing platform (Oxford Nanopore Technologies) was used in these literature, however, the analytical approaches of ONT are different.

Please unify to use Dataset 1, 2, 3, and 4, not to use Table S1 and Table S2. 

Reviewer 2 Report

The application of metagenomics in diverse fields is increasing exponentially including identification and prediction of bacterial pathogens. In this article, the authors evaluated shotgun metagenomics to assess the detection most common enteric bacterial pathogens in the stool. And combined a couple of analytical approaches to assess the pathogen detection using bioinformatics, PCR, sequencing and other techniques.

Figueres are substandard and not currently meeting the journal or readers criteria. Need to improve as suggested in the below minor comments.

  1. Several very short paragraphs can be merged into the main paragraph. For example. Line 111 to 123 can be combined into a single paragraph. Similarly at line 152
  2. Please include the X and Y axis labels for Fig.3
  3. 4 and 5. Wording for the X and Y axis is not readable and barely can see

Reviewer 3 Report

Peterson et al. report an interesting and extremely timely analysis testing the ability to detect known pathogens in stool samples, comparing laboratory techniques and a metagenomics approach. This study is interesting and can help lay the foundations for broader adoption of whole metagenome sequencing in diagnostics environments.

In trying to replicate some of the findings I found a major problem in the methods, that I wish to be addressed. Line #193 lists the used tools but does not give any clue on how to use them. Specifically, it mentions a parameter (Maximum number of mismatches per read for gene) for KAT-SECT while, it is a parameter for SRST2 (--gene_max_mismatch), not to mention that the suggested parameter (150) sounds excessively high. At the same time, KAT-SECT execution must be better annotated, providing the database used.

A the end the sentence "Tables and figures were generated using {list of modules}" is, in my opinion, the equivalent of describing a protocol where "water and salts" are used: I recommend a better explanation.

I would suggest redrawing Figure 1 vertically, allowing for small notes to be added next to each block describing the use of the tool (e.g. "Deconseq: screening for known contaminants"). If a step has a control point (e.g. "if remaining reads are < $THRESHOLD discard the sample"), to summarise the finding and operative suggestions of the paper in the figure.

I recommend the publication of the paper after the methodology part has been improved.

Round 2

Reviewer 1 Report

Thanks for the revision

Reviewer 3 Report

The manuscript improved and is now better describing the overall strategy. While some methods are still not as open as I would recommend (e.g. repository with scripts), the clarity for the reader is adequate.